# Comparison of the Effects of Esketamine/Propofol and Sufentanil/Propofol on the Incidence of Intraoperative Hypoxemia during Bronchoscopy: Protocol for a Randomized, Prospective, Parallel-Group Trial

**DOI:** 10.3390/jcm11154587

**Published:** 2022-08-05

**Authors:** Xiao Huang, Pan Ai, Changwei Wei, Yuan Sun, Anshi Wu

**Affiliations:** 1Department of Anesthesiology, Beijing Chao-Yang Hospital, Capital Medical University, No. 8 Workers’ Stadium South Road, Chaoyang Distinct, Beijing 100020, China; 2Department of Pharmacy, Beijing Chao-Yang Hospital, Capital Medical University, No. 8 Workers’ Stadium South Road, Chaoyang Distinct, Beijing 100020, China

**Keywords:** esketamine, sufentanil, propofol, hypoxemia, bronchoscopy

## Abstract

Background: Propofol, ketamine, and sufentanil are the most commonly used anesthetics during bronchoscopy, alone or in combination, for sedation. Esketamine is an s-enantiomer of ketamine racemate and has both sedative and analgesic effects. Esketamine does not inhibit respiration and maintains hemodynamic stability. This study aims to compare the clinical efficacy of esketamine/propofol with sufentanil/propofol for patients during bronchoscopy. Methods: Patients undergoing bronchoscopy will be randomly assigned to receive either sufentanil/propofol (sufentanil group; *n* = 33; sufentanil: 0.2 μg/kg) or esketamine/propofol (esketamine group; *n* = 33; esketamine: 0.2 mg/kg) for sedation and analgesia. Intraoperative clinical information, general anesthetic drug dosage, the incidence of intraoperative hypoxemia, total time of hypoxemia, awakening time, delirium, nausea and vomiting, adverse reactions, and patient satisfaction will be collected. Discussion: Hypoxia has detrimental effects on patients with respiratory disease. Ameliorating hypoxemia in patients undergoing bronchoscopy is critical. Our results will provide effective sedation with esketamine in patients undergoing bronchoscopy. Trial registration: Chinese clinical trial registry: ChiCTR2200058990.

## 1. Background

Bronchoscopy has been an important tool in the evaluation and management of respiratory disease for many years [1,2,3]. It is a complicated, highly stimulating, painful, and irritating operation performed in patients with pulmonary disease, with a relatively high risk of hypoxemia, coughing, wheezing, and dyspnea [4,5,6]. Intraprocedural sedation, local anesthesia, and general anesthesia have been implemented to avoid such problems in patients undergoing bronchoscopy [7,8,9,10]. Because any movement of the patient could importantly affect the success of the bronchoscopy and because of the great suffering caused to the patient, procedures are usually performed under deep sedation or even general anesthesia.

The combination of propofol and an opioid is a usual anesthesia method during bronchoscopy, despite known side effects such as hypoxemia, postoperative nausea and vomiting, etc., especially for patients without airway devices [11,12]. Joskova et al. found in animal experiments that pharmacodynamic interactions between propofol, sufentanil, and midazolam via GABA_A_ receptor-mediated interactions negatively affect ciliary beat frequency in respiratory epithelial cells [13]. Analgesia and sedation are key for successfully performing bronchoscopy. Opioids may be preferred in combination with other sedative medications to improve patient tolerance during bronchoscopy [14]. In addition to the risk of respiratory depression, opioids have also been reported to cause acute exacerbation of asthma [15]. The sedation used in bronchoscopy requires a rapid onset of action, a short duration of action, rapid recovery, and few adverse effects. Minimizing these risks is therefore an important objective in making sedation surgery safer.

Esketamine, the s-enantiomer of ketamine racemate, has a higher affinity for the NMDA receptor than the r-enantiomer. Hengrui Medicine Co., Ltd. completed the preclinical study of esketamine three years ago and received the clinical study approval from the state food and drug administration [16]. Esketamine combines sedative and analgesic functions. Additionally, its sympathomimetic properties counteract the hemodynamic depression of propofol, thus reducing the risk of cardiovascular and respiratory depression. Due to the increased sympathetic tone, voluntary breathing and airway reflexes are maintained and hypotension is uncommon. Low-dose esketamine not only reduces opioid consumption but also stabilizes breathing, which is evidenced by Jonkman et al. [17]. They showed that esketamine countered respiratory depression by increasing remifentanil-reduced ventilator CO_2_ chemosensitivity.

Esketamine also maintains intraoperative hemodynamic stability. Tu et al. found that esketamine improved hemodynamics, reduced surgical stress and inflammatory response, and promoted postoperative cognitive recovery compared to sufentanil [18]. Esketamine is commonly used for bronchoscopy in children, which is beneficial for patient sedation and reducing the incidence of delirium [19]. Esketamine relaxes bronchiolar muscles and inhibits bronchial constriction due to histamines, thereby reducing tracheal and bronchial muscle spasms.

Endoscopic treatment is now widely used as an effective method to diagnose or treat lung diseases and mainly includes rigid or flexible bronchoscopes. These interventions are high-risk and pose a great challenge to the anesthesiologist. How to establish an appropriate gas exchange to sustain the patient and allow for a smooth operation is an issue to be considered during anesthesia. In addition, reducing perioperative complications, such as intraoperative hypoxemia, airway spasm, and choking, remains an important consideration for the anesthesiologist. Many patients have hypoxia or symptoms of dyspnea before bronchoscopy. In the experience with bronchoscopy at our hospital, we noticed that oxygen saturation (SpO_2_) decreased in many patients who experienced general anesthesia without intubation when undergoing bronchoscopy, despite continuous inspired oxygen.

Hypoxemia is the most frequent complication, and the risk in patients undergoing this procedure is approximately 26–69.1%, due to the different choices of anesthesia and inconsistent indicators for evaluating hypoxemia [12,20,21,22,23,24,25]. Many patients could not tolerate a bronchoscopic procedure without sedation for anxiety and intense stimulation. While the frequently reported regimens include an opioid/propofol combination, other authors [26,27] have suggested that esketamine may be both safe and effective for short surgery. Although previous studies on the safety and validity of esketamine in bronchoscopy are scarce, several studies have demonstrated the effectiveness of ketamine in bronchoscopy, especially for pediatric patients [28,29]. Low-dose intravenous ketamine reduces perioperative opioid use and may provide postoperative analgesia for patients [30]. So far, there is only limited evidence on whether esketamine can decrease the incidence of hypoxemia and improve anesthetic outcomes in patients undergoing bronchoscopy compared with sufentanil or even enhance recovery after surgery.

Data from prospective randomized trials on the effects of esketamine- and sufentanil-assisted general anesthesia on anesthetic awakening are scarce. There is still little evidence of combined esketamine/propofol in bronchoscopy, and it is still open to discussion whether esketamine or sufentanil could reduce the incidence of intraoperative hypoxemia. In our hospital, bronchoscopy has been performed under local anesthesia or sedation with propofol combined with sufentanil by anesthesiologists. On this account, we conducted a randomized controlled trial in this high-risk population to investigate whether opioid-free anesthesia could reduce perioperative hypoxemia. The current study aims to evaluate the efficacy and safety of esketamine/propofol compared with the sufentanil/propofol regimen. We hypothesized that the use of esketamine in patients undergoing flexible bronchoscopy would result in a lower incidence of hypoxemia than standard sufentanil methods.

## 2. Methods/Design

### 2.1. Trial Design

We prepared the study, including ethics approval and pilot testing, which lasted about 3 months. Formal subject recruitment began on 9 March 2022 and is anticipated to be completed by 30 August.

The study is being conducted in Beijing Chao-Yang Hospital, Capital Medical University. The study is designed as a prospective, randomized, controlled, single-trial that includes 66 patients undergoing flexible bronchoscopy and is reported following the Standard Protocol Items: Recommendations for Interventional Trials (SPIRIT) statement [31]. Ethics approval was obtained from the Ethics Committee of the Beijing Chao-Yang Hospital, Capital Medical University on 8 March 2022. The study will be performed following the Declaration of Helsinki [32] and is registered in the clinical trials (22 April 2022, 2022; Clinical Trial: ChiCTR2200058990). Written informed consent is obtained from all participants who meet the inclusion criteria. The present study meets the requirements of the Consolidated Standards for Trial Reporting (CONSORT) checklist. This study is based on a flowchart (Figure 1). Randomization is performed using the Randomization online version software (Table 1).

Potential participants undergoing bronchoscopy are screened for eligibility criteria. All enrolled patients are randomly divided into either the esketamine (Jiangsu Hengrui Pharmaceutical Co., Lianyungang, China) group or the sufentanil (Yichang Humanwell Pharmaceutical Co., Ltd., Yichang, China) group at a 1:1 ratio. We divide the patients into the esketamine group (receives esketamine/propofol sedation) or the sufentanil group (receives propofol/sufentanil sedation) during bronchoscopy. Standard deep sedation with propofol (Fresenius Kabi AB, Rapsgatan 7,751 74 Uppsala, Sweden) target-controlled infusion (TCI) is provided by an anesthesiologist in both groups.

### 2.2. Recruitment and Eligibility Criteria

A special researcher assesses patient eligibility one day before bronchoscopy. The eligible patient, according to the inclusion and exclusion criteria, who agrees to participate in this study is enrolled and randomly assigned to receive either esketamine/propofol- or sufentanil/propofol-based intravenous sedation. The research is canceled in situations where there are patients with serious cardiovascular accidents or who die in the perioperative period.

### 2.3. Inclusion Criteria

Aged 18 years or older at the time of participation.Planning for elective bronchoscopy under general anesthesia without intubation.American Society of Anesthesiologists (ASA) classification I–III.

### 2.4. Exclusion Criteria

Less than 18 years old at the time of participation.SpO_2_ less than 90% after preoperative oxygenation.History of depression or taking antidepressants.History of neuropsychiatric disorders or drug abuse.Inability to cooperate with the scale assessment.Abnormal liver and kidney function.History of hyperthyroidism.Serious adverse reactions such as cardiac arrest and cardiopulmonary resuscitation during bronchoscopy.Severe hypertension, intraocular pressure, or intracranial pressure.A history of allergy to the study drugs or to eggs or soybeans.

### 2.5. Randomization and Blinding

Patients are randomly allocated to the esketamine group or the sufentanil group. A computer-generated randomization schedule is conducted to randomly assign included patients in a 1:1 ratio to reach double-blind treatment with either esketamine or sufentanil. Randomly permuted blocks are used for randomizing and are performed by a trained nurse who specializes in research. The nurse will utilize sealed opaque envelopes to reveal the treatment group on the morning of the operation. Esketamine and sufentanil are both colorless and transparent with 1 mL each. The esketamine or sufentanil is diluted to 10 mL with 0.9% saline by an anesthesia nurse on the day of bronchoscopy. The patients, anesthesiologists, investigators, those assessing the outcomes, and those analyzing the data are blind to the treatment assignment.

### 2.6. Bronchoscopic Procedures

Indication for bronchoscopy includes infection, lung cancer, hemoptysis, interstitial lung disease, and others. The specific type of bronchoscopic procedures is determined by the respiratory surgeon, mainly according to the patient’s diagnosis, preoperative CT scan of the lesion location, size, cancer stage, etc. The types of procedures include inspection with bronchoscopic biopsy, inspection with bronchoalveolar lavage (BAL), bronchial washings, transbronchial lung biopsy (TBLB), conventional transbronchial needle aspiration (TBNA), endobronchial biopsy (EBB), and inspection only.

### 2.7. Anesthetic Management

Bronchoscopy is performed in the outpatient bronchoscopy room. All patients fast from food and water for 8 h before bronchoscopy. Preoxygenation is applied 5 min before anesthesia induction. General anesthesia is performed by anesthesiologists. All patients are monitored by noninvasive blood pressure, electrocardiography, and pulse oximetry and infused with 500 mL NaCl 0.9% at the rate of 250 mL/h. All patients receive oxygen administration (3–6 L/min via a nasal cannula) during the bronchoscopy. After preoxygenation with 100% oxygen, sufentanil (0.2 μg/kg) and propofol (1.5–2.5 mg/kg) are administered during the induction of anesthesia in the sufentanil group or propofol (1.5–2.5 mg/kg) and esketamine (0.2 mg/kg) in the esketamine group. Endoscopic examination starts after the patient has no obvious physical activity and the eyelash reflex has disappeared. Simultaneously, general anesthesia is maintained with propofol at a dose of 4–12 mg kg^−1^ h^−1^. The surgeon injects 5 mL of 2% lidocaine (Tianjin Jinyao Pharmaceutical Co., Tianjin, China) through the tracheoscope into the trachea after the tracheoscope enters the glottis. An amount of 0.5 mg/kg propofol will be added if the swallowing reflex, obvious body motion, or choking response is observed. For patients undergoing transient episodes of SpO_2_ below 90%, high fresh oxygen will be used first to obtain sufficient ventilation and to compensate for airway leaks. If this does not work, we will give the patient mandibular support. If none of these works, we will ask the performer to remove the placed bronchoscope and ventilate the patient for a few minutes until the SpO_2_ rises above 95% and then restart the procedure. In this study, it takes approximately 20 min to perform a bronchoscopy examination, and no additional muscle relaxants or tracheal intubation will be added. Intraoperative vasopressors will be administered to maintain mean arterial blood pressure within 20% of the baseline measurement. Propofol is discontinued at the end of the procedure.

Heart rate (HR), SpO_2_, and noninvasive blood pressure (NIBP) are measured by an independent, blinded observer who collects research data at 3 min intervals. Patients are moved to the postanesthesia care unit (PACU) with oxygen support to maintain the oxygen saturation above 92%. Patients will be transferred to the PACU for Aldrete, delirium, and postoperative nausea and vomiting (PONV) assessment after awakening. The modified Aldrete score is used to assess the recovery from anesthesia and the return of psychotomotoric fitness. This validated assessment consists of 5 aspects (activity, respiration, blood pressure, consciousness, and SpO_2_), with a total score of 10. The clinical data on preoperative diagnosis, type of combined lung disease and blood gas, duration of bronchoscopy, and length of hospital stay are extracted from the medical record.

Safety will be assessed by interview and adverse event (AE) monitoring. All adverse events occurring during the whole study are recorded. Further follow-up is also necessary if there are clinical abnormalities.

### 2.8. Study Outcome

The demographic information and the primary and secondary outcomes are collected and are listed in Table 1. Table 2 provides an overview of the primary and secondary indicators, including a description of the time points for each indicator.

### 2.9. Primary Objective

The primary endpoint of the study—the effect of esketamine versus sufentanil on the incidence of intraoperative hypoxemia—is the rate of SpO_2_ < 90 lasting for the 30 s during the bronchoscopy.

### 2.10. Secondary Objectives

Hemodynamic stability and safety (cardiovascular adverse events such as hypotension or bradycardia, which are measured as a change >25% from baseline and/or necessitating an intervention).

Time from end of bronchoscopy to Aldrete score of 9 (quality of awakening). This is defined as the time for a patient to return to full consciousness from sedation (Aldrete score of 9) after drug administration ceases, which should be accurate on a minute time scale).

Severe intraoperative hypoxemia (SpO_2_  <  75% at any time).

The total duration of intraoperative hypoxemia (total duration of SpO_2_ < 90%).

Incidence and score of PONV at the end of the operation and 1 day after the operation (Likert scale, classified as none, mild, moderate, and severe) [33].

Incidence of delirium within 3 days after bronchoscopy (to the evaluation of a patient’s delirium status and assessment by The Confusion Assessment Method (CAM)).

Incidence of postoperative fever.

Patient satisfaction (one day after the procedure patients will be visited to assess post-procedural satisfaction using the world SIVA International Sedation Task Force Tool [34]. (0 = very unsatisfied, 1 = unsatisfied, 2 = satisfied, 3 = very satisfied).

Propofol dosage in both groups.

The examination time.

The types of procedures.

### 2.11. Sample Size Determination

Due to a lack of data from the literature, we were not able to estimate the differences between the three experimental conditions, which are necessary to calculate the sample size for a within-subject design. The sample size planned for this trial was based on the results of a pilot series preceding this research and on clinical judgment. The required sample size was calculated for the primary outcome and the incidence of perioperative hypoxemia. In our pre-experiment, the incidence of intraoperative hypoxemia was about 64% in the sufentanil group and 30% in the esketamine group. Given the 10% drop-out rate, the randomization of 33 individuals to each treatment group was required to achieve a level of significance and study power of 5% and 80%, respectively [35].

## 3. Statistical Analyses

### 3.1. Baseline Analyses

Continuous variables will be analyzed with the Wilcoxon rank-sum test. Normally distributed data will be assessed by Student’s *t*-test. Categorical data will be assessed using the Fisher’s exact test or χ^2^ test, and risk estimates will be calculated with the odds ratio and 95% confidence intervals (CIs). Data will be presented by mean (standard deviation), median (interquartile range [IQR]), or number (%). Patients with sedation protocol violations or who had serious cardiovascular accidents or who died in the perioperative period or cases that unavoidably required tracheal intubation or had unperformed examinations will be excluded from the final analysis.

### 3.2. Primary Outcome Analysis

The primary outcome of our analysis is the incidence of intraoperative hypoxemia. The comparisons in hypoxemia incidence between the two groups will be evaluated with an χ^2^ test, and 95% CIs will be calculated for the difference in hypoxemia incidence.

### 3.3. Priori Subgroup Analyses

Known confounders have an impact on hypoxemia; therefore, subgroup analysis will be performed by comparing the main results for pre-defined subgroups. These include the preoperative pulmonary function test (mild vs. moderate or severe) and the preoperative partial pressure of oxygen (<80 vs. ≥80 mmHg), to eliminate the impact of these factors on hypoxemia.

### 3.4. Secondary Outcome Analysis

For secondary outcome analyses, the χ^2^ test or the Fisher exact test will be used for categorical variables, including the incidence of intraoperative hypotension or bradycardia, severe intraoperative hypoxemia rate, PONV, delirium, or postoperative fever. The Student’s *t*-test or the nonparametric test will be used for the awakening time, the total duration of intraoperative hypoxemia, and patient satisfaction.

Two-sided *p* values < 0.05 will be considered statistically significant. All statistical analyses will be performed using SPSS Statistics 26.0 and GraphPad Prism 7.

## 4. Discussion

Our trial aims to compare the effect of esketamine/propofol versus sufentanil/propofol on the incidence of hypoxemia in patients undergoing bronchoscopy, thereby providing a better safety and satisfaction plan during bronchoscopy. Given the rise in interventional therapy, bronchoscopy under general anesthesia may become more frequent in the coming future, and this research may provide a safe anesthesia management option for its implementation.

## Figures and Tables

**Figure 1 jcm-11-04587-f001:**
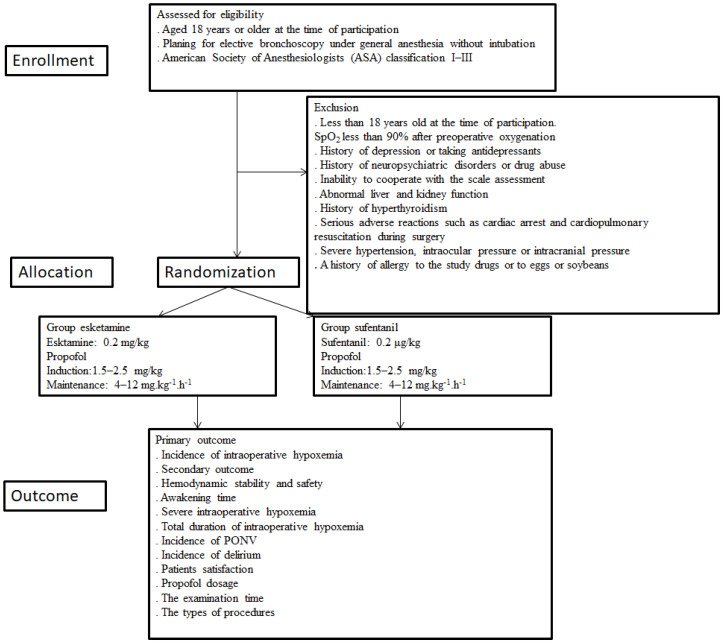
Study enrollment. SpO_2_, oxygen saturation; PONV, postoperative nausea and vomiting.

**Table 1 jcm-11-04587-t001:** Flow chart of the study. PACU, postanesthesia care unit; POD, postoperative day; CAM, the Confusion Assessment Method; PONV, postoperative nausea and vomiting. “×” means perfoming in that time.

Items	Visit 1	Visit 2	Visit 3	Visit 4	Visit 5
Time	One day before bronchoscoy	Morning of bronchoscopy	During bronchoscopy	PACU	POD1
Eligiblility screen	×				
Medical history and medication history	×				
Surgery history	×				
Physical examination	×				
Inclusion/exclusion criteria	×	×			
Informed consent		×			
Randomization		×			
Demographics			×	×	
Hemodynamic variables			×	×	
Study drug			×	×	
Outcome indicators			×	×	×
CAM				×	×
Aldrete				×	
PONV				×	×
Adverse events			×	×	×
Patient satisfaction					×

**Table 2 jcm-11-04587-t002:** Definitions of the multiple outcomes included in this study protocol. SpO_2_, oxygen saturation; PONV, postoperative nausea and vomiting.

Outcome	Definition	Instrumental/Tool
Incidence of intraoperative hypoxemia	The rate of SpO_2_ < 90% lasts for 30 s during the bronchoscopy.	SpO_2_
Hemodynamic stability and safety	Cardiovascular adverse events such as hypotension or bradycardia, which will be measured as a change >25% from baseline and/or necessitating an intervention.	Non-invasive blood pressure and electrocardiogram
Awakening time	Time from end of bronchoscopy to Aldrete score of 9.	Aldrete
Severe intraoperative hypoxemia	SpO_2_ < 75% at any time during the bronchoscopy.	SpO_2_
Total duration of intraoperative hypoxemia	Total duration of SpO_2_ < 90% during the bronchoscopy.	SpO_2_
PONV	The severity of PONV (none, mild, moderate, and severe).	Likert scale
Delirium	To evaluation of a patient’s delirium status within 3 days after bronchoscopy.	The Confusion Assessment Method
Adverse events	Accidental or adverse reactions to sedatives that threaten or cause patient injury or discomfort	World SIVA International Sedation Task Force Tool
Patient satisfaction	Patient satisfaction with the procedure experience (very unsatisfied, unsatisfied, satisfied, very satisfied)	Likert scale

## Data Availability

Not applicable.

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
