# Peer review of "Comparison of the Effects of Esketamine/Propofol and Sufentanil/Propofol on the Incidence of Intraoperative Hypoxemia during Bronchoscopy: Protocol for a Randomized, Prospective, Parallel-Group Trial"

_jcm, 2022, doi:10.3390/jcm11154587_

Round 1
Reviewer 1 Report
This is a protocol article for a randomized controlled trial to evaluate the efficacy and safety of sedation with sufentanil/propofol and esketamine/propofol during flexible bronchoscopy. No similar study has been conducted in bronchoscopy, making it very interesting and of broad interest to the reader.
However, the manuscript submitted was difficult to peer review because Figure 1 was not included, line and page numbers were not attached, and spelling errors were noticeable (e.g., 2 in SpO2 is not in subscript, ml is correctly in mL, no single space in front of the unit, etc.).
In the section on sample size determination, in their pre-experiment, was the incidence of intraoperative hypoxemia really about 64% in the esketamine group and about 30% in the sufentanil group?
I would suggest a re-review of this manuscript after these corrections are made.
Author Response
We thank the reviewer for pointing out these issues. These comments are all valuable and very helpful for improving our paper. We have carefully considered the suggestion of Reviewer 1 and made some changes.
We uploaded Figure 1 as a separate file in the first submission. We apologize that you did not see Figure 1 during peer review due to our mistakes. We have included Figure 1 in the revised manuscript.
We apologize for the informality of our manuscript. Line and page numbers have been attached in the revised manuscript.
We have many spelling errors in the manuscript, and we apologize for our carelessness. We agree with this suggestion and have modified terminology throughout the text as appropriate. Spelling errors have been corrected in the revised manuscript. We work on the manuscript for a long time and the repeated addition and removal of sentences and sections led to poor readability. We hope that the revised manuscript has been substantially improved.
In our pre-experiment, 11 patients were included in the sufentanil group and 10 patients in the esketamine group, and hypoxemia occurred in 7 people in the sufentanil group, with an incidence of 64%. Hypoxemia occurred in 3 people in the esketamine group, with an incidence of 30%. Given the 10% drop-out rate, randomization of 33 individuals to each treatment group was required to achieve a level of significance and study power of 5% and 80%, respectively.

Reviewer 2 Report
Manuscript presents study protocol. Authors aim to compare two models of
intravenous anesthesia: with use of Esketamine + Propofol and Sufentanil + Propofol.
Authors assume that use of Esketamine might lower risk of incidental hypoxemia during bronchoscopy. Authors say that use of sufentanil is the standard method. I cannot agree with this.
Authors stated that combination of propofol and an opiod has become the preferred choice during bronchoscopy but there is only one position which supports this statement.
Why did authors choose Esketamine + Propofol and Sufentanil + Propofol. No data in references about Esketamine and its use in treatment. Not necessarily in anesthesiology. Why Esketamine and not Ketamine?
During reading of the manuscript a couple of questions arose:
1. Is this rigid or flexible bronchoscopy?
2. Do patients receive premedication?
3. Is Lidocaine being used?
4. Why 8 hours of starvation?
5. When is preoxygenation being applied?
6. What are advantages and disadvantages of using Esketamine and Sufentanil?
7. Hang (51) and Yin (48) described ketamine, not esketamine.
8. What does it mean "in previous surgical expierence, we notice...". Where and when? Lack of references.
9. No definiton of hypoxemia.
10. In the Manuscript I was provided with there is no Figure 1.
11. Research hypothesis is repeated in Methods.
12. Inclusion and exclusion criteria should be numbered for better reading of the text.
13. Is it actually correct to define bronchoscopy as a surgery?
14. In my opinion, ASA III patients should not be qualified for the research
15. Methods of administration are not defined. IV way is not obvious to everybody. It is not said whether drugs were administered with use of a dosing pump or in fractionated doses. If there was a pump, what kind and what manufacturer was it?
16. Manufacturer of drugs is not mentioned anywhere.
17. Were propofol and esketamine and sufentail administered via same cannula?
18. What does "3-6 flow nasal cannula" mean?
19. It was not stated in what situations research has been cancelled.
20. Propofol is supposed to be administered in a dose between 4 and 12 mg. What is the reason of such range of doses and what criteria were used?
21. Why is propofol dosage not compared in both groups?
22. What vasopressor should be used?
23. Table 2. - What does FFB stand for?
24. Table 2. - Table is unreadable, wrongly edited.
25. In Discussion, aim of paper is repeated for the third time.
26. In a study protocol-type paper Discussion should not be placed. One could not just compare the results to different than published. All necessary information should be mention in the background. Important Conclusion is the one in which authors are able to include information that gives ground for taking up described research.
Author Response
Thank you for your decision and constructive comments on my manuscript. We are very grateful to Reviewer 2 for reviewing the paper so carefully. And we have learned a lot from them. We have tried our best to improve the manuscript and have modified some inappropriate expressions. Please check them. Revision notes, point-to-point, are given as follows:
Manuscript presents study protocol. Authors aim to compare two models of
intravenous anesthesia: with use of Esketamine + Propofol and Sufentanil + Propofol.
Authors assume that use of Esketamine might lower risk of incidental hypoxemia during bronchoscopy. Authors say that use of sufentanil is the standard method. I cannot agree with this.
Authors stated that combination of propofol and an opiod has become the preferred choice during bronchoscopy but there is only one position which supports this statement.
Why did authors choose Esketamine + Propofol and Sufentanil + Propofol. No data in references about Esketamine and its use in treatment. Not necessarily in anesthesiology. Why Esketamine and not Ketamine?
Response: Thanks for your comments. We have changed our arbitrary statement that the use of sufentanil is the standard method. We agree that the use of sufentanil is not the standard method accepted by all. In our hospital, the combination of propofol and an opioid has become the preferred choice during bronchoscopy. The reason we choose Esketamine + Propofol and Sufentanil + Propofol are listed as follows: firstly esketamine is newer than ketamine and therefore more likely to stimulate research interest. Secondly, we are limited by the fact that our hospital currently only has esketamine, not ketamine. Finally, because there is much research on ketamine in all kinds of surgeries, we are more interested in observing the usage of esketamine in bronchoscopy.
Point 1: Is this rigid or flexible bronchoscopy?
Response 1: Thanks for your comments. This is flexible bronchoscopy.
Point 2: Do patients receive premedication?
Response 2: Thanks for your comments. Patients don’t receive premedication.
Point 3: Is Lidocaine being used?
Response 3: Thanks for your comments. The surgeon injects 5 ml of 2% lidocaine through the tracheoscope into the trachea after the tracheoscope enters the glottis.
Point 4: Why 8 hours of starvation?
Response 4: Thanks for your comments. Routine fasting for 8 hours before general anesthesia is required. This is to prevent nausea and vomiting, regurgitation and aspiration even aspiration pneumonia during general anesthesia, which will threaten the safety of patients.
Point 5: When is preoxygenation being applied?
Response 5: Thanks for your comments. Preoxygenation is applied 5 minutes before anesthesia induction. Adequate preoxygenation increases the patient's ability to tolerate hypoxia during induction of anesthesia.
Point 6: What are advantages and disadvantages of using Esketamine and Sufentanil?
Response 6: Thanks for your comments. Esketamine is a new drug that has both sedative and analgesic effects. Since it has been marketed for a short time, the clinical effects still need to be verified and explored. It is contraindicated in patients with severe hypertension and coronary artery disease, cardiac insufficiency, severe pulmonary hypertension, and high cranial or intraocular pressure. And patients with epilepsy, psychiatric history, hyperthyroidism, and adrenal pheochromocytoma showed used with caution. Sufentanil is a commonly used opioid analgesic with a strong analgesic effect. It has been in clinical use for many years and has a wide range of applications. But it has many side effects such as nausea and vomiting and hypoxemia.
Point 7: Hang (51) and Yin (48) described ketamine, not esketamine.
Response 7: Thanks for your comments. We have modified the reference in the revised manuscript.
Point 8: What does it mean "in previous surgical expierence, we notice...". Where and when? Lack of references.
Response 8: Thanks for your comments. We want to express that “in the experience with bronchoscopy at our hospital (Beijing Chao-Yang Hospital, Capital Medical University)”. We have corrected this sentence in the revised manuscript. Because our presentation is based on our clinical work experience, there are no references.
Point 9: No definiton of hypoxemia.
Response 9: Thanks for your comments. We define hypoxemia as SpO2<90 last for the 30s during the bronchoscopy in the “Primary objective” section. The definition of hypoxemia is based on the previous literature (as follows), the short duration of the bronchoscopy procedure and the tolerance time for hypoxia in our clinical experience.
References:
Xie J, Covassin N, Fan Z, Singh P, Gao W, Li G, Kara T, Somers VK. Association Between Hypoxemia and Mortality in Patients With COVID-19. Mayo Clin Proc. 2020 Jun;95(6):1138-1147. doi: 10.1016/j.mayocp.2020.04.006.
Templeton TW, Miller SA, Lee LK, Kheterpal S, Mathis MR, Goenaga-Díaz EJ, Templeton LB, Saha AK; Multicenter Perioperative Outcomes Group Investigators. Hypoxemia in Young Children Undergoing One-lung Ventilation: A Retrospective Cohort Study. Anesthesiology. 2021 Nov 1;135(5):842-853. doi: 10.1097/ALN.0000000000003971.
Preciado P, Tapia Silva LM, Ye X, Zhang H, Wang Y, Waguespack P, Kooman JP, Kotanko P. Arterial oxygen saturation and hypoxemia in hemodialysis patients with COVID-19. Clin Kidney J. 2021 Feb 1;14(4):1222-1228. doi: 10.1093/ckj/sfab019.
Turan A, Essber H, Saasouh W, Hovsepyan K, Makarova N, Ayad S, Cohen B, Ruetzler K, Soliman LM, Maheshwari K, Yang D, Mascha EJ, Ali Sakr Esa W, Kessler H, Delaney CP, Sessler DI; FACTOR Study Group. Effect of Intravenous Acetaminophen on Postoperative Hypoxemia After Abdominal Surgery: The FACTOR Randomized Clinical Trial. JAMA. 2020 Jul 28;324(4):350-358. doi: 10.1001/jama.2020.10009.
Point 10: In the Manuscript I was provided with there is no Figure 1.
Response 10: Thanks for your comments. We have provided Figure 1 in the revised manuscript.
Point 11: Research hypothesis is repeated in Methods.
Response 11: Thanks for your comments. We have removed the repeated research hypothesis in the revised manuscript.
Point 12: Inclusion and exclusion criteria should be numbered for better reading of the text.
Response 12: Thanks for your comments. We have numbered the inclusion and exclusion criteria in the revised manuscript.
Point 13: Is it actually correct to define bronchoscopy as a surgery?
Response 13: Thanks for your comments. Bronchoscopy is not suitable to be defined as a surgery. We have corrected the expression of “surgery” as bronchoscopy in the revised manuscript.
Point 14: In my opinion, ASA III patients should not be qualified for the research.
Response 14: Thanks for your comments. Patients undergoing bronchoscopy at our hospital is mainly ASA 2-3. Many previous studies have focused on the effects of esketamine in patients with ASA1-2. There is a lack of studies in patients with ASA3. Based on previous studies, we presume that esketamine is suitable for patients with ASA grade 3 theoretically. In the statistical analysis after the completion of the study, we could compare between patients in ASA class 3 with other patients to observe the effect of esketamine and sufentanil on hypoxemia.
Point 15: Methods of administration are not defined. IV way is not obvious to everybody. It is not said whether drugs were administered with use of a dosing pump or in fractionated doses. If there was a pump, what kind and what manufacturer was it?
Response 15: Thanks for your comments. For each patient, esketamine, sufentanil, and propofol induction are administered one-time intravenously. Anesthesia maintenance: propofol is administered with use of a dosing pump (Beijing Fresenius Kabi Pharmaceutical Co.).
Point 16: Manufacturer of drugs is not mentioned anywhere.
Response 16: Thanks for your comments.
Lidocaine (Tianjin Jinyao Pharmaceutical Co. China)
Propofol(Fresenius Kabi AB
Rapsgatan 7,751 74 Uppsala
Sweden)
Sufentanil(Yichang humanwell pharmaceutical CO., LTD, Hubei, China)
Esketamine(Jiangsu Hengrui Pharmaceutical Co., LTD Jangsu, China)
Dosing pump(Beijing Fresenius Kabi Pharmaceutical Co., LTD)
Point 17: Were propofol and esketamine and sufentail administered via same cannula.
Response 17: Thanks for your comments. Propofol, esketamine and sufentail are administered via same cannula.
Point18: What does "3-6 flow nasal cannula" mean?
Response 18: Thanks for your comments. We have modify this sentences as “All patients receive oxygen administration (3-6 L/min via a nasal cannula) during the surgery”.
Point 19: It was not stated in what situations research has been cancelled.
Response 19: Thanks for your comments. The research will be canceled in situations where patients with serious cardiovascular accidents, or die in the perioperative period. We have added this in the “Recruitment and eligibility criteria” section in the revised manuscript.
Point 20: Propofol is supposed to be administered in a dose between 4 and 12 mg. What is the reason of such range of doses and what criteria were used?
Response 20: Thanks for your comments. General anesthesia is maintained with propofol at a dose of 4-12mg.kg-1.h-1. The reason for this range of doses and criteria were determined by the range specified in the drug's instructions.
Point 21: Why is propofol dosage not compared in both groups?
Response 21: Thanks for your comments. The propofol dosage will be compared in both groups. We have added this item in the “study outcome section” in the revised manuscript.
Point 22: What vasopressor should be used?
Response 22: Thanks for your comments. We use norepinephrine, epinephrine or ephedrine based on the patient's condition and vital signs monitoring.
Point 23: Table 2. - What does FFB stand for?
Response 23: Thanks for your comments. FFB in Table 2 stands for “Flexible fiberoptic bronchoscopy”. We have modified this to bronchoscopy in the revised manuscript.
Point 24: Table 2. - Table is unreadable, wrongly edited.
Response 24: Thanks for your comments. Table 2 has been edited again in the revised manuscript.
Point 25: In Discussion, aim of paper is repeated for the third time.
Response 25: Thanks for your comments. We have removed the repeated aim of paper in Discussion.
Point 26: In a study protocol-type paper Discussion should not be placed. One could not just compare the results to different than published. All necessary information should be mention in the background. Important Conclusion is the one in which authors are able to include information that gives ground for taking up described research.
Response 26: Thanks for your comments. We apologize for the confusion generated by the previous version of the manuscript. In previous study protocol-type papers, discussion have been placed. We have placed the necessary information in the background in the revised manuscript.

Round 2
Reviewer 1 Report
I greatly appreciate your revisions to my comments. However, there were many points that remained uncorrected, therefore, I will point them out below.
Major comments
Please clarify the stratification factors used for allocation. In particular, the examination time and patient invasiveness vary greatly depending on the bronchoscopic procedure. In my opinion, the types of procedures should be allocated equally.
The regulations regarding bronchoscopic procedures should be described in more detail.
Is it acceptable to examine multiple target lesions and multiple procedures (e.g., systematic staging with EBUS-TBNA and rEBUS-TBB for peripheral lesions) simultaneously?
What cases are excluded from the analysis?
E.g., sedation protocol violations, unavoidably required tracheal intubation cases, unperformed examinations.
Minor comments
In our pre-experiment, 11 patients were included in the sufentanil group and 10 patients in the esketamine group, and hypoxemia occurred in 7 people in the sufentanil group, with an incidence of 64%. Hypoxemia occurred in 3 people in the esketamine group, with an incidence of 30%. Given the 10% drop-out rate, randomization of 33 individuals to each treatment group was required to achieve a level of significance and study power of 5% and 80%, respectively.
Lines 245
Therefore, it is correct to replace the sufentanil group with the esketamine group.
Lines 68, 84, 151, 188, 191, 197, 204, 217, 227, 228, 282, Table 1
I repeat, please use subscripts.
Lines 22, 23, 179, 181, 183, 217, Figure 1, Table 2
Similarly, please insert a single-byte space.
0.2 μg/kg
0.2 mg/kg
esketamine/propofol (esketamine group;
0.2 μg/kg
0.2 mg/kg
4-12 mg.kg-1.h-1.
30 s
Lines 197
Please spell out the following abbreviations.
HR
NIBP
Lines 44
The citation is inappropriate. Reference 11 does not seem to contain such a mention. Please cite the appropriate reference.
Lines 149
A history of allergy to the study drugs or to eggs or soybeans should be added to the exclusion criteria.
I will appreciate it if you accept all suggested corrections, and resubmit your revised manuscript.
Author Response
To reviewer 1
Thank you for your carefully reviewing of this manuscript and constructive suggestions, which are very useful to us. We have learned a lot from them. We have tried our best to answer and revise our manuscript as the comments required. We have modified some inappropriate expressions. Please check them. The red part in the revised manuscript has been revised according to the comments. Thanks again for your diligent contribution to this manuscript.
Major comments
Please clarify the stratification factors used for allocation. In particular, the examination time and patient invasiveness vary greatly depending on the bronchoscopic procedure. In my opinion, the types of procedures should be allocated equally.
Reply: Thanks for your comments. The study is double-blind randomized controlled, so the types of procedures is supposed to be allocated equally in the two group. Our study included patients undergoing general anesthesia without intubation, so we only included the common types of bronchoscopy (excluding some complex or new techniques that require intubation). The types of procedures include inspection with bronchoscopic biopsy, inspection with bronchoalveolar lavage (BAL), bronchial washings, transbronchial lung biopsy (TBLB), conventional transbronchial needle aspiration (TBNA), endobronchial biopsy (EBB) and inspection only (this type is relatively uncommon in our hospital). We have added this to the “bronchoscopic procedures”in the revised manuscript. (Line 172-179) We will compare the examination time and the types of procedures in the final statistics. We have added this to the “Study outcome- Secondary objectives”in the revised manuscript. (Line 247-248)
The regulations regarding bronchoscopic procedures should be described in more detail.
Reply: Thanks for your comments. Indication for bronchoscopy include infection, lung cancer, hemoptysis, interstitial lung disease and others. The specific type of bronchoscopic procedures is determined by the respiratory surgeon, mainly according to the patient's diagnosis, preoperative CT scan of the lesion location, size, cancer stage, etc. We have added this to the “bronchoscopic procedures”in the revised manuscript. (Line 172-179)
Is it acceptable to examine multiple target lesions and multiple procedures (e.g., systematic staging with EBUS-TBNA and rEBUS-TBB for peripheral lesions) simultaneously?
Reply: Thanks for your comments. It is acceptable to examine multiple target lesions and multiple procedures simultaneously in our study.
What cases are excluded from the analysis?
E.g., sedation protocol violations, unavoidably required tracheal intubation cases, unperformed examinations.
Reply: Thanks for your comments. Patients with sedation protocol violations, serious cardiovascular accidents or die in the perioperative period, unavoidably required tracheal intubation cases or unperformed examinations will be excluded from the final analysis. We have added this regulations in the “Statistical Analyses BaseLine Analyses” section in the revised manuscript. (Line 266-269)
Minor comments
In our pre-experiment, 11 patients were included in the sufentanil group and 10 patients in the esketamine group, and hypoxemia occurred in 7 people in the sufentanil group, with an incidence of 64%. Hypoxemia occurred in 3 people in the esketamine group, with an incidence of 30%. Given the 10% drop-out rate, randomization of 33 individuals to each treatment group was required to achieve a level of significance and study power of 5% and 80%, respectively.
Lines 245
Therefore, it is correct to replace the sufentanil group with the esketamine group.
Reply: Thanks for your comments. We have replaced the sufentanil group with the esketamine group in the revised manuscript. (Line 256)
Lines 68, 84, 151, 188, 191, 197, 204, 217, 227, 228, 282, Table 1
I repeat, please use subscripts.
Reply: Thanks for your comments. We apologize for the misspelling of SpO2 due to our stupidity and carelessness in the first revised manuscript. We have used subscripts in the revised manuscript. (Line 68, 84, 151, 197, 199, 206, 213, 226, 236, 237, 296, 453, 458, Table 1 and Figure 1)
Lines 22, 23, 179, 181, 183, 217, Figure 1, Table 2
Similarly, please insert a single-byte space.
0.2 μg/kg
0.2 mg/kg
esketamine/propofol (esketamine group;
0.2 μg/kg
0.2 mg/kg
4-12 mg.kg-1.h-1.
30 s
Reply: Thanks for your comments. We have inserted a single-byte space in the revised manuscript. (Line 22, 23, 188, 190, 192, 226, Figure 1 and Table 2)
Lines 197
Please spell out the following abbreviations.
HR
NIBP
Reply: Thanks for your comments. We have spelled out the abbreviations of HR and NIBP in the revised manuscript. (Line 206)
Lines 44
The citation is inappropriate. Reference 11 does not seem to contain such a mention. Please cite the appropriate reference.
Reply: Thanks for your comments. We have changed the reference as the followings: (Line 44)
Gunathilaka PKG., Jat KR., Sankar J., Lodha R., Kabra SK. Propofol versus Fentanyl for Sedation in Pediatric Bronchoscopy: A Randomized Controlled Trial. Indian Pediatr. 2019;56(12):1011-1016.
Zha B., Wu Z., Xie P., Xiong H., Xu L., Wei H. Supraglottic jet oxygenation and ventilation reduces desaturation during bronchoscopy under moderate to deep sedation with propofol and remifentanil: A randomised controlled clinical trial. Eur J Anaesthesiol. 2021;38(3):294-301. doi: 10.1097/EJA.0000000000001401.
Lines 149
A history of allergy to the study drugs or to eggs or soybeans should be added to the exclusion criteria.
Reply: Thanks for your comments. We have added a history of allergy to the study drugs or to eggs or soybeans in the exclusion criteria in the revised manuscript (Line 160)

Reviewer 2 Report
I have no further comments.
Author Response
Thank you very much.